# High Tartronic Acid Content Germplasms Screening of Cucumber and Its Response to Exogenous Agents

**DOI:** 10.3390/foods13101484

**Published:** 2024-05-10

**Authors:** Zhongren Zhang, Yixin Qu, Ruijia Wang, Yaru Wang, Songlin Yang, Lei Sun, Sen Li, Yiming Gao, Yuming Dong, Xingwang Liu, Huazhong Ren

**Affiliations:** 1College of Horticulture, China Agricultural University, Beijing 100193, China; zzr9906@cau.edu.cn (Z.Z.); sy20223173016@cau.edu.cn (Y.Q.); wangruijia2023@163.com (R.W.); yangsonglin@cau.edu.cn (S.Y.); s20193172452@cau.edu.cn (L.S.); BS20233171035@cau.edu.cn (S.L.); gaoyiming@cau.edu.cn (Y.G.); dym@cau.edu.cn (Y.D.); renhuazhong@cau.edu.cn (H.R.); 2Sanya Institute, China Agricultural University, Sanya 572025, China

**Keywords:** tartronic acid, cucumber, harvesting period, fruit parts, microbial agent

## Abstract

Tartronic acid is known for its potential to inhibit sugar-to-lipid conversion in the human body, leading to weight loss and fat reduction. This compound is predominantly found in cucumbers and other cucurbit crops. Therefore, cultivating cucumbers with high tartronic acid content holds significant health implications. In this study, we assessed the tartronic acid content in 52 cucumber germplasms with favorable overall traits and identified 8 cucumber germplasms with elevated tartronic acid levels. Our investigation into factors influencing cucumber tartronic acid revealed a decrease in content with fruit development from the day of flowering. Furthermore, tartronic acid content was higher in early-harvested fruits compared to late-harvested ones, with the rear part of the fruit exhibiting significantly higher content than other parts. Foliar spraying of microbial agents increased tartronic acid content by 84.4%. This study provides valuable resources for breeding high tartronic acid cucumbers and offers practical insights for optimizing cucumber production practices.

## 1. Introduction

Cucumber (*Cucumis sativus* L.), an annual climbing herb of the Cucurbitaceae family, holds significant economic importance worldwide [1]. Recognized for its fragrant and delectable fruit, cucumber is primarily consumed in its fresh form. With the continuous improvement of living standards and a growing emphasis on health, consumers are increasingly interested in the nutritional quality and health benefits of vegetable products. Cucumber boasts a rich composition of carotenoids, vitamins, minerals, and a variety of organic acids. Among these, tartronic acid stands out for its documented weight-loss and lipid-lowering effects due to its ability to inhibit the conversion of sugar into fat [2,3,4,5]. In 2016, cardiovascular diseases (CVDs) emerged as the leading cause of death among Chinese adults. Effective control of blood fat levels has proven instrumental in reducing the morbidity and mortality associated with CVDs [6,7]. Therefore, the breeding of cucumbers with an elevated content of tartronic acid holds promise in making a distinctive contribution to national health protection within the realm of horticultural crops.

Tartronic acid, characterized by the molecular formula C_3_H_4_O_5_, structural formula HOOCCH(OH)COOH, and a molecular weight of 120.06, is a small organic acid. Its inhibitory effects on carbohydrate-to-fat conversion were first discovered in the 1950s when it was observed that tartronic acid or its sodium salt could impede this process [8]. In vivo injection tests conducted on mice revealed that tartronic acid increased liver glycogen, reduced fat content, and resulted in weight loss. This provides further evidence of tartronic acid’s function to inhibit the conversion of sugar into fat, thereby controlling the body weight of animals [9]. Studies have also demonstrated that tartronic acid can reduce body fat in *Caenorhabditis elegans* without affecting their movement and swallowing behaviors [10]. The underlying mechanism of tartronic acid involves acting as an inhibitor of malic enzyme, a key enzyme in pyruvate metabolism. By inhibiting the conversion of glucose into pyruvate in the glycolytic pathway, tartronic acid suppresses fat synthesis [11]. Importantly, humans cannot synthesize tartronic acid and must obtain it from their diet. Cucumber, rich in organic tartronic acid, serves as an excellent dietary source.

The exploration of naturally synthesized tartronic acid has primarily focused on a limited number of cucurbit crops, notably cucumber and winter melon, with a relatively superficial understanding. The research emphasis was mainly on the optimization of extraction techniques and content determination, while the genetic mechanisms and metabolic pathways remain unexplored. In the flesh of winter melon, high-performance liquid chromatography–tandem mass spectrometry (HPLC-MS/MS) identified five organic acids with a relative content exceeding 0.5%, including tartronic acid at 1.15% [12]. The extraction method for determining the tartronic acid content using HPLC in the wax gourd variety chieh-qua was optimized, serving as a foundation for its extraction and application in that crop [13]. Early studies on cucumber fruit utilized HPLC for tartronic acid determination. However, in recent years, a novel method based on ion chromatography has been introduced. This method boasts a short operation time, high precision and accuracy, excellent reproducibility, and immunity to interference from other impurities. This methodology has been selected for the current study [14,15].

The content of tartronic acid in plants is influenced by several factors. Different cucumber germplasm resources exhibit varying levels of tartronic acid. A comprehensive analysis of 56 cucumber germplasms from four ecotypes revealed a 5.7-fold difference between the highest and lowest levels [16], highlighting the significant impact of germplasm type on tartronic acid content. The application of nitrogen fertilizer has been identified as a factor that can increase the content of tartronic acid in cucumber [17,18]. Additionally, the external application of organic acid, such as malic acid, has demonstrated a noteworthy improvement in cucumber physiological characteristics [19]. Foliar spraying of exogenous sucrose has also been reported to improve the resistance of cucurbit crops, such as melon and cucumber, at the seedling stage [20,21]. Meanwhile, microbial fertilizer plays a positive role in promoting plant growth and improving fruit quality [22]. The application of microbial fertilizer not only improves dry matter quality, sugar content, and vitamin C content of cucumbers but also positively influences cucumber flavor [23,24]. However, despite these findings, the impact of treatment with either organic acids or microbial fertilizers as exogenous agents on the content of cucumber tartronic acid remains unexplored.

In this study, tartronic acid content in 52 cucumber germplasms was determined using ion chromatography, and those with a high content exceeding 0.5 mg/g were selectively identified. The investigation further delved into the influences of the fruit development period, harvesting timing, fruit part, and exogenous agent treatments on cucumber tartronic acid content. These findings offer a reference for further research on the metabolism and efficient production of cucumber tartronic acid. The outcomes of this study lay a crucial foundation for the subsequent selection and breeding of cucumber varieties characterized by high tartronic acid content, paving the way for their high-quality commercial production. The implications of this research extend to the broader goal of advancing our understanding of cucumber tartronic acid metabolism for practical applications in agriculture.

## 2. Materials and Methods

### 2.1. Plant Materials and Growth Conditions

The laboratory created a collection of 52 cucumber germplasm resources, showcasing excellent comprehensive traits. This assortment comprises 31 North China-type cucumbers, 21 European greenhouse-type cucumbers, and two hybrids: 19024 (3631-1 × 3611-12) and 19114 (5569-9 × 3595-2). These materials were planted in the spring of 2022 in the sunlight greenhouse of Dongbeiwang Agricultural Base, Haidian District, Beijing, China. All management was carried out in accordance with the basic field production and was consistently applied according to the standard production requirements in the field. The cucumber germplasms used in this study are shown in Table 1.

### 2.2. Exogenous Agent Treatment

Exogenous agents include sucrose, malic acid, citric acid, and microbial bacterial agent 1 (trade name: Carbon’s Treasure, main ingredient: *Bacillus sphaericus*, purchased from Baolai-Leelai Shandong Baolai-Leelai Bio-Tech Co., Ltd. (Taian, China). Dilute the masterbatch at a ratio of 1:300).

The experiment comprised four treatment groups: Treatment 1 (T1), sprayed with a 0.50% sucrose solution; Treatment 2 (T2), sprayed with a 0.05% malic acid solution; Treatment 3 (T3), sprayed with a 0.33% microbial bacterial agent 1; and Treatment 4 (T4), sprayed with a 0.04% citric acid solution. The control group (CK) was sprayed with water. Exogenous treatments commenced early in the fruiting stage, with three replications for each treatment. Spraying occurred once every five days, totaling five applications. The foliar spraying method was employed, ensuring even coverage of the entire plant until the leaf surface slightly dripped. All other conditions remained consistent across treatments except for the varied spraying reagents. The type, concentration, and treatment of exogenous agents are determined based on actual production practices and previous work experience, with due consideration given to safety concerns.

### 2.3. Determination of Tartronic Acid Content

For reproducibility, only well-developed fruits at nodes 8–16 of the plant were selected for the experiment. The fruits were juiced and the extracted paste underwent centrifugation (12,000 rpm, 8 min). The supernatant was then passed through an activated CNWBOND LC-C18 solid phase extraction column. Activation involved a sequential process: 2 mL of methanol, followed by 2 mL of a 1:1 methanol–water solution, then 2 mL of water. After 2 min, 2 mL of water was added, and the activation was allowed to proceed for 30 min, followed by discarding 1 mL. The remaining portion was passed through a 0.22 μm membrane, and the filtrate was used for detection. The content was determined using ion chromatography. The ion chromatograph used was a Thermo Scientific Diongx ICS-5000+. For anion analysis, the analytical column employed was a Dionex IonPAC AS11-HC (250 mm × 4 mm), with a guard column of IonPac AG11-HC (4 mm × 50 mm). The column temperature was maintained at 30 °C, and the eluent solution consisted of KOH solution at a flow rate of 1 mL/min. The standard curve was generated based on standard samples.

### 2.4. Statistical Analysis

The SPSS 26.0 software was used for statistical analysis. To determine the significant difference, Student’s *t* test and one-way analysis of variance (ANOVA) with Bonferroni’s test were employed. A *p*-value of less than 0.05 was considered significant. Graphs were generated using GraphPad Prism 8.0.2.

## 3. Result

### 3.1. Significant Variation in Tartronic Acid Content among Different Cucumber Germplasms

Initially, tartronic acid content was analyzed in 21 European greenhouse-type cucumber germplasm materials (Table 2). The results showed that the content of tartronic acid ranged from 0.09 to 0.76 mg/g, with an average of 0.44 mg/g. The content in 2125 was the lowest at 0.09 mg/g, and the content in 6101-11 was the highest at 0.76 mg/g.

Furthermore, an analysis of fruit shape, fruit color, melon tumor size, melon tumor density, fruit ribs, and melon gloss was conducted. Attempts were made to establish connections between tartronic acid content and observable appearance qualities, but no correlations were identified.

The tartronic acid content in 31 cucumber germplasm materials of the North China type was determined (Table 3). The range of tartronic acid content spanned from 0.06 to 0.81 g/mg, with an average content of 0.28 mg/g. The lowest content was observed in 3631-1 at 0.06 mg/g, while the highest was recorded in 3569-16 at 0.81 mg/g.

Additionally, an analysis of cucumber appearance quality was conducted. However, no correlation was identified between tartronic acid content and cucumber appearance quality.

In conclusion, significant variations were observed in the tartronic acid content among different germplasm materials, with no correlation found between tartronic acid content and appearance quality.

Furthermore, we identified eight cucumber germplasm materials with tartronic acid content higher than 0.50 mg/g, using twice the tartronic acid content of commercially available cucumber varieties as the base standard (Appendix A). Among these, three were North China-type germplasm materials (3569-16, 5667, and 5634-1), and five were European greenhouse-type germplasm materials (2079-10, 6101-5, 2073-1, 2128, and 6101-11).

### 3.2. Influence of Fruit Developmental Stages on Tartronic Acid Content

To investigate the impact of different stages of fruit development on tartronic acid content, two North China-type germplasms 3611-1 and 3611-2 were used as test materials, and the tartronic acid content was determined on the day of anthesis and 3, 6, 9, and 12 days after anthesis (DAA) (Figure 1). The general trend of tartronic acid content exhibited a decline as the fruit underwent growth and development, with the highest content observed on the day of flowering and a significant decrease in tartronic acid content from 6 DAA onward. Further analysis showed that from the day of anthesis to 12 DAA, the tartronic acid content in 3611-1 fruit decreased from 1.37 mg/g to 0.66 mg/g, while in 3611-2 fruit, it decreased from 1.05 mg/g to 0.33 mg/g. These findings underscore the influence of the fruit development period on tartronic acid content, with the highest levels observed on the day of flowering, followed by a consistent decrease during growth and development.

### 3.3. The Impact of Harvesting Period on Tartronic Acid Content in Fruits

To assess the influence of the harvesting period on tartronic acid content, measurements were taken at early and late harvest periods (30 and 60 days after planting). The results showed that during the early harvest period, the tartronic acid content of 19024 and 19114 was 0.55 mg/g and 0.68 mg/g, respectively. In the late harvest period, the tartronic acid content of 19024 and 19114 was 0.34 mg/g and 0.32 mg/g, respectively (Figure 2). Upon further analysis, it was found that the tartronic acid content in the early harvesting period was 61.76% higher for 19024 and 112.50% higher for 19114 compared to the late harvesting period. This suggests that 19114 was more influenced by the growth period of the plant. In conclusion, the results showed that tartronic acid content in cucumber fruits was higher in the early harvest period compared to the late harvest period of the plant.

### 3.4. The Tartronic Acid Content in Different Parts of the Fruit

The commercial fruit from germplasm 19024 was evenly divided into four parts: “top of fruit”, “middle of fruit”, “end of fruit”, and “fruit handle”. Equal-weight samples were taken from each part for testing (Figure 3). Comparing the different fruit parts, the content of tartronic acid was highest in the “end of fruit” at 0.81 mg/g, followed by “fruit handle” at 0.46 mg/g, “middle of fruit” at 0.43 mg/g, and the lowest in the “top of fruit” at 0.25 mg/g. The content of the mixed sample of commercial fruits of 19024 was 0.55 mg/g (Table 2). Specifically, the content in the “end of fruit” part was 47.27% higher than the mixed sample, while the “top of fruit” part was 54.55% lower than the mixed sample. The “middle of fruit” and “fruit handle” parts showed relatively similar content. In summary, the results underscore significant variation in tartronic acid content across different parts of the fruit, with the “end of fruit” exhibiting significantly higher content compared to other parts.

### 3.5. Influence of Exogenous Agents on the Tartronic Acid Content of Cucumber Fruits

To investigate the impact of exogenous agents on tartronic acid content, the exogenous treatment was initiated from the early stage of plant development, and the resulting tartronic acid content was measured. Tartronic acid content under control (CK), sucrose treatment (T1), malic acid treatment (T2), microbial agent 1 treatment (T3), and citric acid treatment (T4) were 0.45 mg/g, 0.50 mg/g, 0.52 mg/g, 0.83 mg/g, and 0.40 mg/g, respectively (Figure 4).

Compared to the control, the exogenous treatments of sucrose (T1) and malic acid (T2) increased the tartronic acid content by 11.1% and 15.6%, respectively, but did not reach a significant level. However, the microbial treatment (T3) treatment significantly increased the tartronic acid content by 84.4% compared to the control.

In conclusion, the results demonstrate that exogenous agents can impact cucumber tartronic acid content, with the T3 treatment (microbial agent 1) significantly increasing the tartronic acid content.

## 4. Discussion

Dyslipidemia is a well-recognized risk factor for cardiovascular disease morbidity and mortality, and in China, the prevalence of dyslipidemia in the population aged 18 years and older reached 40.4% in 2012 [25,26,27]. Effective control of dyslipidemia is crucial for maintaining public health. Tartronic acid, a natural component unique to cucumber, inhibits the conversion of sugar compounds into fat in the human body, preventing fat accumulation. It demonstrates significant therapeutic effects on weight loss and the prevention of coronary heart disease and other illnesses. The existing commercially available cucumber varieties generally have low tartronic acid content, typically below 0.3 mg/g (Appendix A). Screening of cucumber germplasm materials with high tartronic acid content is essential for improving the lipid-lowering effect of cucumber fruits. In this study, we identified eight cucumber germplasms with tartronic acid content exceeding 0.5 mg/g, which was more than double that of market varieties, among 31 North China-type cucumbers and 21 European greenhouse-type cucumbers using ion chromatography. Meanwhile, we also thoroughly observed and recorded the fruit appearance characteristics of 52 materials, and preliminarily found that the cucumber’s tartronic acid content was not necessarily related to its ecological type and appearance quality. In future cucumber breeding work, there is potential to achieve both excellent appearance quality and high tartronic acid content.

The nutrient composition of fruits undergoes dynamic changes, with organic acids being a crucial factor influencing fruit quality. In apples, malic acid dominates, with its content increasing 2–4 weeks after flowering and subsequently decreasing until fruit ripening [28]. In tomato fruits, citric acid content increases during ripening and reaches a maximum at the post-harvest stage, while malic acid concentration decreases during ripening and maturation. A similar trend was observed in bell pepper fruits [29,30]. In this study, tartronic acid content in cucumbers was determined at 0, 3, 6, 9, and 12 DAA. Results showed the highest content on the day of flowering, followed by a decreasing trend with the growth and development of fruits. We hypothesized that tartronic acid in the fruit was mainly synthesized during ovary development.

The harvesting period of horticultural products is often closely related to fruit quality. In Kuril balsam pear, hardness, titratable acid, and chlorophyll contents were highest during the early harvest period, while Vitamin C content peaked during the mid-harvest period [31]. Similarly, in this study, cucumber fruit tartronic acid content was found to be higher in the early harvest compared to the late harvest. Substance contents in different parts of horticultural product fruits also exhibit variations. For jujube, measurements during different growth periods revealed higher soluble sugar content in the fruit shoulder compared to the fruit top [32]. Likewise, a comparison of sugar and acid contents in different parts of red heart dragon fruit indicated higher levels in the fruit apex, the surrounding area, and the bottom part of the fruit [33]. In this study, cucumber fruits were divided into four parts for the determination of tartronic acid content, revealing significant differences among parts. The “end of fruit” part showed notably higher content, and, interestingly, the fruit handle also exhibited relatively high tartronic acid levels.

With the development of agriculture, prolonged and excessive use of chemical fertilizers and pesticides has led to soil structure degradation, reduced soil activity, environmental pollution, biodiversity decline, and potential threats to human health [34,35]. Microbial fungicides, emerging as sustainable fertilizers, offer eco-friendly and green alternatives that can partially replace chemical fertilizers, mitigating environmental risks while enhancing crop yield and quality [24,36,37]. In this study, exogenous preparations were applied through foliar spraying, revealing an 84.4% increase in tartronic acid content compared to the control under the treatment of “microbial bacterial agent 1”. This finding demonstrates the significant potential of microbial agents in enhancing cucumber tartronic acid content. It not only establishes the groundwork for cucumber agricultural production with high tartronic acid content but also presents novel ideas for the sustainable development of green agriculture. Research on tartronic acid is still in its early stages, primarily focused on optimizing extraction and assay in a few cucurbit crops. The complete picture of biosynthetic pathways, regulatory mechanisms, and key genes involved is yet to be elucidated. We propose that the active ingredients or metabolites of microbial agents might influence the gene expression profile of these key pathways, subsequently affecting the tartronic acid content in cucumber fruits.

In our study, we measured the tartronic acid content of 52 materials and investigated the influencing factors to clarify the variation of tartronic acid content, which provided material and data support for the cloning of related genes and genetic improvement based on molecular means. Meanwhile, in the face of the fact that the breeding improvement work has not yet been completed, we also proposed a method to rapidly and efficiently increase the tartronic acid content of cucumber, i.e., the treatment of exogenous microbial agents. In the future, biochemistry, genetics, and molecular biology should be used to further analyze the mechanism of cucumber tartronic acid synthesis, metabolism, and regulation.

## 5. Conclusions

The aim of our research is to identify germplasm resources suitable for breeding high tartronic acid cucumbers and provide technical support for their production. In this study, we determined the tartronic acid content in 52 different germplasm resources in our laboratory and identified 8 germplasms with tartronic acid content exceeding 0.50 mg/g. Three of these germplasms belonged to the North China type, labeled 3569-16, 5667, 5634-1, while five belonged to the European greenhouse type, labeled 2079-10, 6101-5, 2073-1, 2128, and 6101-11. Their tartronic acid content was two times more than that of the commercially available varieties.

Our investigation into influencing factors revealed a decreasing trend in tartronic acid content from the day of anthesis to 12 DAA. Fruits harvested earlier had significantly higher levels compared to those harvested later. Additionally, the “end of fruit” part exhibited significantly higher content than other sections of the same fruits. Notably, exogenous application of microbial agents rapidly and efficiently increased the tartronic acid content of cucumber fruits by 84.4%.

## Figures and Tables

**Figure 1 foods-13-01484-f001:**
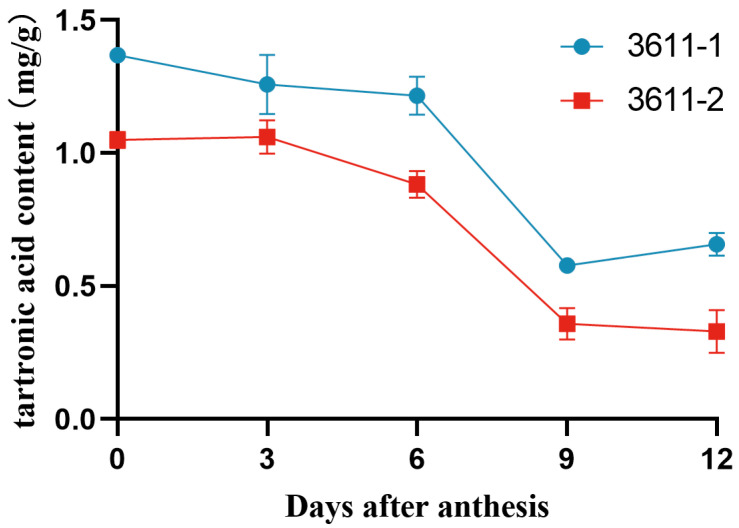
Tartronic acid content in cucumber fruits at different developmental stages.

**Figure 2 foods-13-01484-f002:**
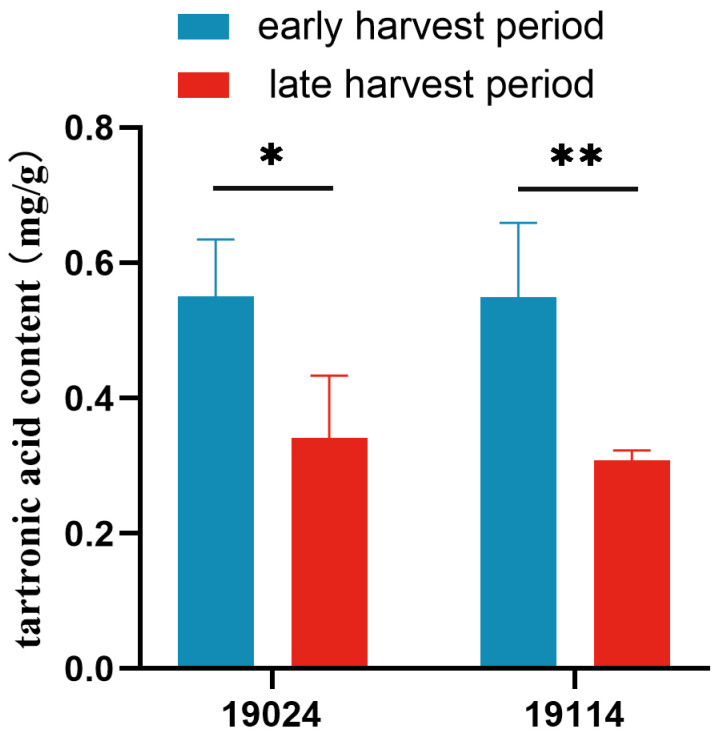
Tartronic acid content in cucumber fruits at different harvest periods. * indicates *p* < 0.05, ** indicates *p* < 0.01.

**Figure 3 foods-13-01484-f003:**
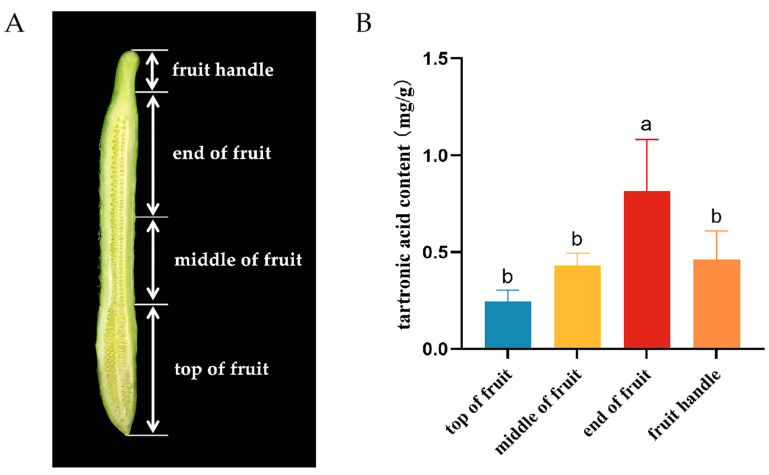
Tartronic acid content in different fruit parts. (**A**) Different parts of fruit; (**B**) Content of tartronic acid in different parts of fruit. Standard deviations are represented by vertical bars. Values labeled with different letters indicate significant variations (*p* < 0.05).

**Figure 4 foods-13-01484-f004:**
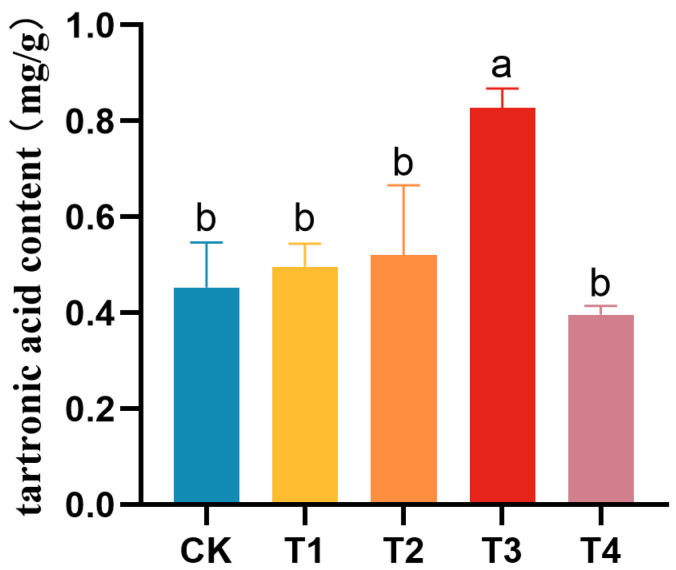
Tartronic acid content in cucumber fruits treated with various exogenous agents. Standard deviations are represented by vertical bars. Values labeled with different letters indicate significant variations (*p* < 0.05).

**Table 1 foods-13-01484-t001:** List of cucumber germplasm materials.

North China Type	North China Type	North China Type	European Greenhouse Type	European Greenhouse Type
3631-1	3667	3599-2	2125	2127
5569-9	3661	3599-1	2073-2	2091
3577	3658	5553-1	2089-11	2086-2
3554-6	3595-2	3611-1	2126	2109
3541-1	3630-1	3660	2119	2098
3611-2	3542-1	3569-8	2079-1	2079-10
3634-1	3628-1	5634-1	2105	6101-5
3549-1	5552-1	5667	2102	2073-1
3595-6	3594-1	3569-16	6101-4	2128
3548-1	3610-4		2103	6101-11
3467-1	5594-5		2113	

**Table 2 foods-13-01484-t002:** Tartronic acid content and appearance quality in European greenhouse types.

Serial Number	Content (mg/g)	Fruit Shape	Fruit Color	Melon Tumor Size	Melon Tumor Density	Fruit Ribbing	Melon Gloss
2125	0.09 ± 0.01	round	Green	Small	Sparse	no fruit ribbing	light
2073-2	0.11 ± 0.00	Short cylinder	Dark Green	None	None	no fruit ribbing	glossy
2089-11	0.28 ± 0.10	Short cylinder	Green	None	None	no fruit ribbing	light
2126	0.32 ± 0.03	Short cylinder	Green	None	None	no fruit ribbing	light
2119	0.40 ± 0.09	Long cylinder	Yellow	Small	Sparse	tiny fruit ribbing	light
2079-1	0.40 ± 0.02	round	Green	Small	Sparse	no fruit ribbing	light
2105	0.40 ± 0.05	Long cylinder	Yellow–Green	Small	Sparse	tiny fruit ribbing	light
2102	0.41 ± 0.08	round	Green	Small	Middle	no fruit ribbing	light
6101-4	0.41 ± 0.01	Short cylinder	Green	None	None	no fruit ribbing	light
2103	0.43 ± 0.04	Short cylinder	Dark Green	None	None	no fruit ribbing	light
2113	0.46 ± 0.04	Short cylinder	Green	Small	Sparse	no fruit ribbing	dull
2127	0.47 ± 0.02	Short cylinder	Green	None	None	no fruit ribbing	light
2091	0.48 ± 0.10	Long cylinder	White	Small	Sparse	tiny fruit ribbing	light
2086-2	0.48 ± 0.04	Short cylinder	Green	None	None	no fruit ribbing	light
2109	0.48 ± 0.03	Short cylinder	Green	Small	Sparse	no fruit ribbing	dull
2098	0.49 ± 0.00	Long cylinder	Green	Middle	Sparse	tiny fruit ribbing	light
2079-10	0.54 ± 0.05	round	Yellow	None	None	no fruit ribbing	light
6101-5	0.55 ± 0.04	Short cylinder	Yellow–Green	None	None	no fruit ribbing	light
2073-1	0.59 ± 0.15	Long cylinder	Green	None	None	no fruit ribbing	light
2128	0.63 ± 0.09	Short cylinder	Green	None	None	no fruit ribbing	light
6101-11	0.76 ± 0.06	Long cylinder	Yellow–Green	None	None	no fruit ribbing	light
Average Content	0.44						

**Table 3 foods-13-01484-t003:** Tartronic acid content and appearance quality in North China types.

Serial Number	Content (mg/g)	Fruit Shape	Fruit Color	Melon Tumor Size	Melon Tumor Density	Fruit Ribbing	Melon Gloss
3631-1	0.06 ± 0.01	Elongate	Green	Small	Middle	Scanty fruit ribbing	glossy
5569-9	0.07 ± 0.01	Elongate	Green	Middle	Middle	Scanty fruit ribbing	light
3577	0.07 ± 0.01	Elongate	Green	Middle	Middle	Scanty fruit ribbing	light
3554-6	0.08 ± 0.01	Elongate	Green	Middle	Middle	Scanty fruit ribbing	light
3541-1	0.10 ± 0.02	Elongate	Green	Middle	Middle	Scanty fruit ribbing	light
3611-2	0.11 ± 0.03	Elongate	Green	Middle	Middle	Scanty fruit ribbing	light
3634-1	0.17 ± 0.01	Elongate	Green	Middle	Middle	Scanty fruit ribbing	light
3549-1	0.17 ± 0.01	Elongate	Green	Middle	Middle	Scanty fruit ribbing	dull
3595-6	0.18 ± 0.07	Elongate	Green	Small	Middle	Scanty fruit ribbing	glossy
3548-1	0.18 ± 0.02	Elongate	Green	Middle	Middle	Scanty fruit ribbing	light
3467-1	0.19 ± 0.06	Elongate	Green	Middle	Middle	Scanty fruit ribbing	light
3667	0.20 ± 0.01	Elongate	Green	Middle	Middle	Scanty fruit ribbing	light
3661	0.20 ± 0.04	Elongate	Green	Middle	Middle	Scanty fruit ribbing	light
3658	0.21 ± 0.01	Elongate	Green	Middle	Middle	Scanty fruit ribbing	light
3595-2	0.21 ± 0.07	Elongate	Green	Middle	Middle	distinct fruit ribbing	light
3630-1	0.23 ± 0.07	Elongate	Green	Middle	Middle	distinct fruit ribbing	light
3542-1	0.24 ± 0.08	Elongate	Green	Middle	Middle	Scanty fruit ribbing	light
3628-1	0.27 ± 0.11	Elongate	Green	Middle	Middle	Scanty fruit ribbing	light
5552-1	0.29 ± 0.01	Elongate	Green	Middle	Middle	Scanty fruit ribbing	light
3594-1	0.31 ± 0.08	Elongate	Green	Middle	Middle	Scanty fruit ribbing	light
3610-4	0.31 ± 0.04	Elongate	Green	Middle	Middle	Scanty fruit ribbing	light
5594-5	0.32 ± 0.02	Elongate	Green	Middle	Middle	distinct fruit ribbing	light
3599-2	0.32 ± 0.16	Elongate	Green	Middle	Middle	distinct fruit ribbing	light
3599-1	0.33 ± 0.03	Elongate	Green	Middle	Middle	Scanty fruit ribbing	light
5553-1	0.39 ± 0.08	Elongate	Green	Middle	Middle	Scanty fruit ribbing	light
3611-1	0.45 ± 0.16	Elongate	Green	Middle	Middle	distinct fruit ribbing	light
3660	0.48 ± 0.16	Elongate	Green	Middle	Middle	distinct fruit ribbing	light
3569-8	0.49 ± 0.14	Elongate	Green	Middle	Middle	Scanty fruit ribbing	light
5634-1	0.61 ± 0.20	Elongate	Green	Middle	Middle	distinct fruit ribbing	light
5667	0.76 ± 0.22	Elongate	Green	Middle	Middle	distinct fruit ribbing	light
3569-16	0.81 ± 0.13	Elongate	Green	Middle	Middle	Scanty fruit ribbing	light
Average Content	0.28						

## Data Availability

The original contributions presented in the study are included in the article/Appendix A, further inquiries can be directed to the corresponding author.

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
