# Peer review of "High Tartronic Acid Content Germplasms Screening of Cucumber and Its Response to Exogenous Agents"

_foods, 2024, doi:10.3390/foods13101484_

Round 1
Reviewer 1 Report
Comments and Suggestions for Authors
Paper analyses the tartaric acid content in different cucumbers cultivated in China and Europe. The tartaric acid content is related to different variants of the fruit, the plant and t the growing process.
The paper is analytically interesting, however, the positive finding of the research is not apparent.
Furthermore, the conclusions report a summary of what has been done, but there are no real conclusions and practical feedback on the data obtained.
Paper may be considered for Foods journal, but only after reviewing the conclusions and the practical aim of the research.
Reviewer 2 Report
Comments and Suggestions for Authors
Dear Mr. Noppawut Pongboon
Assistant Editor
Foods
Please find enclosed my revision of the manuscript ID: foods-2991819
Manuscript title:
High Tartronic Acid Content Germplasms Screening of Cucumber and 2 Its Response to Exogenous Agents
The manuscript seems to be interesting in the field of healthy and functional foods and may be suitable in our journals with some minor revisions including
1- Please indicate the reasons for selecting each concentration of exogenous agents
2- Please indicate more information about microbial agents including its concentration and application methods
3- Please add more information about tartronic acid determination to be more repeatable.
4- I don't understand how to use Duncan's multiple-range tests. Why did you use this test? The reason why is not clear to me. Many statisticians pointed out the problem in this test: Type I error rate is high (i.e. Ryan 1959; Scheffe 1959; Petrinovich and Hardyck 1969…….
5- In each sub-experiment, you used different germplasm, please indicate why you used the same germplasm in all experiments.
6- The discussion is very poor and need to improved by introducing the role of external agents in increasing tartronic acid as well as indicate the correlation between fruit characteristics and its content of Tartronic acid.
Yours truly

Comments on the Quality of English Language
Minor editing of English language required
Reviewer 3 Report
Comments and Suggestions for Authors
Article title:
“High Tartronic Acid Content Germplasms Screening of Cucumber and Its Response to Exogenous Agents".
Comments:
Abstract: Reformulated again, mentioning the treatments, highlighting the goal and the most important results.
Introduction: Write in detail the role of the studied exogenous agents and their relationship to tartronic acid in the fruits of the Cucurbitaceae family
Materials and Methods
- Line 105: Bacillus sphaericus (Italic). Also, added the total count of it.
- Added the growth conditions in the pot experiment (soil analysis type and their analysis)
- What about the different concentrations of exogenous agents used? Why these concentrations?
Results
- 3.1. Section: What about the interaction between exogenous agents and cucumber germplasm (Table 2,3). - Line 137-138: Added this text in MM section, and clear in results section. - 3.4. Section: Clear in MM section. - Fig. (3A), It appears clearer in the picture. - What about the effect of exogenous agents on the different parts of the fruit.
Discussion
It is almost identical to the introduction, which lacks the role of exogenous agents used and their effect is the presence of tartronic acid in the fruits of Cucumber.
Conclusion
- and identified eight germplasms with tartronic acid content exceeding (clear)
Round 2
Reviewer 3 Report
Comments and Suggestions for Authors The manuscript is accepted in the present form.